# It Had to Be You: Carl Schmitt on Exclusion and Political Reasoning

Andrés Rosler [1,2]

1  Facultad de Filosofía y Letras, Universidad de Buenos Aires (UBA), Buenos Aires 1406, Argentina; andres.rosler@gmail.com
2  Consejo Nacional de Investigaciones Científicas y Técnicas (CONICET), Buenos Aires 1917, Argentina

**Abstract:** In this paper, I would like to tackle first Schmitt's defence of the role of exclusion in political reasoning and his attendant rejection of extreme political pluralism. I shall then move on to explain not only why there is nothing Nazi—or even antisemitic—about Schmitt's concept of the political, but rather the other way around: Schmitt's concept of the political not only must have been used against National Socialism but it did not fail to have his fair share of Jewish, or at the very least Zionist, enthusiasts.

**Keywords:** Carl Schmitt; the political; political reasoning; exclusion



## 1. Introduction

A main leitmotif of Carl Schmitt's concept of the political—and perhaps of his entire work—is that, unlike moral reasoning that makes a point of treating all people in equal and universal terms, political reasoning cannot afford to be all-inclusive since political inclusion can only be achieved at the expense of the exclusion of alternative political ideas and institutional designs, and much the same deal applies to territories and human beings themselves.

In Schmitt's eyes, there is no way around political exclusion; the only question is how it will take place. Naturally, in this day and age, when it is only the praises of inclusion that are sung, to say that Schmitt's theory of the autonomy of politics is not an easy sell is quite an understatement. What is more, there are scholars that still tie up the concept of the political—particularly its exclusionary penchant—to National Socialism, somewhat understandably in the light of Schmitt's own decision to collaborate with Hitler's regime mainly between 1933 and 1936. However, Schmitt's decision to do so cannot be reconciled with his own concept of the political. Indeed, the exclusionary nature of political reasoning is actually proven by the need of liberal democracies to exclude precisely National Socialism (among others).

In what follows, I shall tackle first Schmitt's defence of the role of exclusion in political reasoning and his attendant rejection of extreme political pluralism. The first part of the paper starts off with the distinction between enemies and criminals and the way it has been misunderstood as an invitation to kill people or even worse. Schmitt's distinction is actually meant to keep violence and conflict to a minimum and this is why his concept of the political can be understood as a second-order ethic, a kind of political morality. This is followed by a discussion of Schmitt's distinction between different types of enmity, the way Schmitt's favourite conception of enmity (the one espoused by the *jus publicum europaeum*) managed to keep violence relatively contained, a rather brief survey of the main jurists and philosophers that illustrate Schmitt's approach to exclusion, and the way this disposition allowed for the emergence of the distinction between state and civil society with its attendant, rather moderate, pluralism.

The second part of the paper takes its cue precisely from the way in which pluralism in its extremest form would be tantamount to an open-house policy even for the enemies of

pluralism like National Socialism, thereby showing how, in Schmitt's eyes, the very exclusionary nature of political reasoning requires us to take the enemies of liberal democracy very seriously. Since, time and again, Schmitt's concept of the political has been taken to embody antisemitism, the third and last section on the German Zionist reception of the concept of the political illustrates why there is nothing structurally antisemitic, let alone National Socialist, in it[1].

## 2. Enemies and Criminals

Perhaps the most important normative point of Schmitt's concept of the political is its emphasis on the distinction between enemy and criminal. This is one of the main reasons why the autonomy of the political can be described as a second-order or political ethic that attempts to provide a "politics stripped of self-righteous moralizing", in which the enemy is "treated not as a criminal to be banished, but simply as an enemy to be overcome. This was indeed a minimal ethic: not love your neighbour, but respect your enemy". According to Schmitt, the political is not "identified with the sheer intensification of the struggle against an enemy" as it happened with the outbreak of modern revolutions, "but with the limit which keeps this intensification within bounds, a reciprocity in which the enemy is recognized as legitimate, and respected as an enemy—that is to say, not just in his harmless, non-political status as a fellow human being" [1] (pp. 107–109). Thus, Schmitt's concept of the political entails the very opposite of the annihilation of political enemies.

However, not a few readers of Schmitt's very well-known essay have missed this aspect of his theory of political reasoning. For instance, in his pioneering and otherwise rather perceptive review of *The Concept of the Political*, Helmut Kuhn claims that "for the romantic Schmitt, the state of nature is an ideal that must be reestablished through politics". This is why, according to Kuhn, Schmitt "is an inverted Rousseau: the shepherd's idyll—with some overstatement—has become a predator's idyll" [2] (p. 194). Kuhn thus seems to believe that Schmitt is a fanatic of conflict, or what is worse, predation.

To show how wrong this belief is, we should simply recall that "Schmitt's dictum that the political is 'the distinction of friend and enemy' was formulated simultaneously with the *Verfassungslehre* (1928) and is directly related to it. What *Der Begriff des Politischen* understands as a problem, *Verfassungslehre* attempts to resolve, relating the political to the constitutional" [3] (p. 95). Actually, it would be inaccurate to say even that *The Concept of the Political* is only occupied with conflict or the problem, whereas *Constitutional Theory* deals with order or the solution, since the concept of the political itself dwells in both dissociation and *association*, enmity and *friendship*, exclusion and *inclusion*. In fact, originally, the first version of *The Concept of the Political* was thought of as a chapter on "Conceptual determination of the political" for *Constitutional Theory* [4] (p. 129, n. 694)[2].

The exclusionary nature of the political, viz. of the distinction between friend and enemy or the boundary that includes and excludes—or that includes by exclusion—may be moderate, medium, or extreme. The moderate or intellectual shape of the political comes in the form of the polemical nature of political concepts. What has been recently ascribed to the Cambridge School of the history of political thought was already part of the concept of the political as understood by Schmitt:

> All political concepts, images, and terms have a polemical meaning. They are focused on a specific conflict and are bound to a concrete situation; the result (which manifests itself in war or revolution) is a friend–enemy grouping, and they turn into empty ghostlike abstractions when this situation disappears. Words such as state, republic, society, class, as well as sovereignty, constitutional state, absolutism, dictatorship, economic planning, neutral or total state, and so on, are incomprehensible if one does not know exactly who is affected, combated, refuted, or negated by such a term. [5] (pp. 30–31)

The medium shape of the political appears in what can be described as the citizenship principle. At the very root of a political community, there is a formative exclusion that functions as the principle of citizenship: by the very fact of saying that X is a citizen of

political community Y, we are saying not only that X is a member of Y (and hence defining a group inclusively, in terms of who its members are), but also thereby excluding some people from membership in that group.

Finally, the extreme version of the political shows up in the form of outright enmity or hostility, so much so that physical damage is inflicted upon other people, viz. political violence may well be brought to bear upon the conflict.

To be sure, Schmitt does not rule out the possibility that the right decision, what is politically correct, may well be not to engage in any violent action, "The definition of the political suggested here neither favors war nor militarism, neither imperialism nor pacifism" [5] (p. 33). Nevertheless, if it comes to that, we must make up our minds about what kind of enmity we shall favour.

Schmitt distinguishes between three kinds of enmity: conventional, real, or absolute[3]. Conventional enmity is the kind that managed to achieve "The Overcoming of Civil War by War in State-Form", "the rationalization and humanization of war, i.e., the possibility of bracketing war in international law". This is the conception of enmity defended by Schmitt in *The Concept of the Political*. Since humans tend to understand their enemies as evil and ugly [5] (p. 27), this is "an incredibly human accomplishment, that men disclaimed a discrimination and denigration of the enemy" [7] (p. 90).

The bracketing of war by the *jus publicum europaeum* was made possible by the fact that "the problem of just war had been divorced from the problem of *justa causa*, and had become determined by formal juridical categories" [8] (pp. 140–141). Religious and civil wars were turned into a "war in form": "Only in this way, only by limiting war to conflicts between territorially defined European states, could a conflict between these spatially defined units be conceived of as *personae publicae* [public persons] living on common European soil and belonging to the same European 'family'. Thus, it was possible for each side to recognize the other as *justi hostes*. Thereby, war became somewhat analogous to a duel...". Hence, "European soil became the theatre of war (*theatrum belli*), the enclosed space in which politically authorized and militarily organized states could test their strength against one another under the watchful eyes of all European sovereigns". Both parties to a war were considered as belligerents, they "had the same political character and the same rights; both recognized each other as states. As a result, it was possible to distinguish an enemy from a criminal. Not only was the concept of enemy able to assume a legal form, but the enemy ceased to be someone 'who must be annihilated'" [8] (pp. 141–142).

The second kind of enmity is what Schmitt dubs "real" enemy. In *The Concept of the Political*, this type of enmity is opposed to the notion employed by just war theory: "To demand of a politically united people that it wage war for a just cause only is either something self-evident, if it means that war can be risked only against a real enemy...". It is in this context that Schmitt explains that political violence is autonomous vis-à-vis criminal violence: "If there are really are enemies in the existential sense as meant here, then it is justified, but only politically, to repel and fight them psychically" [5] (p. 49)[4].

"The Real Enemy" is also the title of a section in the last chapter of *The Theory of the Partisan*, which is meant to be an *Intermediate Commentary on The Concept of the Political*—as stated in the subtitle of the book. At this juncture, Schmitt describes real enmity in the following terms: "An enemy is not someone who, for some reason or other, must be eliminated and destroyed because he has no value. The enemy is on the same level as am I. For this reason, I must fight him to the same extent and within the same bounds as he fights me, in order to be consistent with the definition of the real enemy by which he defines me". In Schmitt's view, the true partisan was the French republican General Raoul Salan, who took a fundamentally defensive position and thus "presupposes a fundamental limitation of enmity. The real enemy will not be declared to be an absolute enemy, also not the last enemy of mankind" [7] (p. 85).

Speaking of absolute enmity, we have now entered the realm of the third and last kind of enmity, the one totally incompatible with the concept of the political. This is the achievement of revolutionary partisanship as understood, e.g., by Lenin, who "as a

professional revolutionary of global civil war, went still further [than Clausewitz] and turned the real enemy into an absolute enemy" [7] (p. 93). This type of enmity provokes wars "unusually intense and inhuman because, by transcending the limits of the political framework, it simultaneously degrades the enemy into moral and other categories", and therefore our own enemy becomes *eo ipso* an enemy of mankind, "a monster that must not only be defeated but also utterly destroyed. In other words, he is an enemy who no longer must be compelled to retreat into his borders only" [5] (p. 36).

Thus, if from a purely descriptive standpoint, the concept of the state presupposes the concept of the political—as stated at the inception of Schmitt's essay on the political, from a truly normative standpoint, it is the other way around: the state understood as a token of political unity has priority over the political, and hence, paraphrasing Larry David, Schmitt's slogan is curb your political enthusiasm. If we are lucky, to the extent that the concepts of the state and the political can be identified, the state, understood as a neutral sphere between parties—let alone factions—will call the political shots[5]. This is why Schmitt understands the state "as the model of political unity, as the bearer of the most astonishing of all monopolies, viz. the monopoly of political decision, this work of art of European form and Western rationalism" [6] (p. 10).

Back at the height of the *jus publicum europeaum*, the state meant real progress in terms of the containment of war since the autonomy of the political achieved a relativization of enmity, "a great progress in the sense of humanity. Certainly, it is not easy to achieve it, since it is very hard for human beings not to take their enemy for a criminal. In no case it is a progress in the sense of humanity to outlaw as reactionary and criminal the contained war of European international law, and instead to unleash revolutionary class- or race-enmities in the name of just wars, not to be able to distinguish between enemy and criminal anymore, and not to be willing to do so anymore" [6] (pp. 11–12). The job started by the revolution in 1789 was completed at the beginning of First World War I when, having rejected international law, "Europe stumbled into a world war that dethroned the old world from the center of the earth and destroyed the bracketing of war it had created" [8] (p. 239).

It goes without saying that the state as the monopoly of political decision not only relativized external war and enmity, but it also did away with internal or civil war by providing a neutral sphere where members of different religions, ideologies, etc., were able to come together. Initially, this was the work of political jurists, such as Jean Bodin, "a typical *politique*" that stood for the state as the "higher and neutral unity" of France during the religious civil wars [6] (p. 10).

Near the end of the XVI century in France, the name *politique* was something that people were called rather than call themselves, and it indicated a mixed state between Huguenot and "true Catholic" [9] (p. 176), viz. the mixed state of those who argued for the autonomy of political reasoning vis-à-vis religion. *Politiques* were not said to be "true Catholics" because they were an influential group of moderate Catholics that, by the 1560s, had come to the conclusion that any attempt to impose a policy of religious uniformity by force would constitute, at the very least, a serious tactical mistake. The so-called party of *politiques*, Catholics as they were, "argued that uniformity was no longer worth preserving, however valuable it might be in itself, if the cost of enforcing it seemed liable to be the destruction of the commonwealth" [10] (pp. 249–251).

These jurists were called *politiques* because they were thought to have been influenced by Machiavelli's *The Prince* and its emphasis on political stability at the expense of religion[6], and thus their neutrality was taken to be atheism. For instance, the Catholic pamphleteer Artur Desiré said of Michel de L'Hospital, Chancellor of France, that: "His virtue is being a Proteus, / His neutrality is being an atheist [*Sa vertu d'est d'estre un Prothée,/Sa neutralité d'estre athée*]" [12] (p. 191). However, L'Hospital's point was that there was no need to be a "true Catholic" to be a citizen; all it took was to abide by the laws of the land, "It is not a question of establishing religion, but of establishing the republic. And many can be citizens that will not be Christians" [13] (p. 173).

This is why, in reference to Bodin's characteristic equilibrated attitude and pursuit of a common ground, Schmitt holds "That such an equilibrated attitude in a time of fanatical wars of religion and partisan politics is called 'political', belongs in a particularly instructive chapter of the history of the concept of the political" [14] (p. 202).

Thomas Hobbes is a truly *politique* philosopher who took a leaf out of Bodin's book: *what* we have legal reason to do must be a function of *whom* we believe to be a source of legally authoritative reasons for action (*quis judicabit?*). Hobbes transforms then the question of what is truly legal into the question of whose judgment we ought to take as authoritative. Whereas "ADVICE is an *instruction* or *precept* [praeceptum] in which the reason for following it is drawn *from the matter itself*", "COMMAND is an *instruction* in which the reason for following it is drawn from *the will of the instructor*". Hobbes then adds that since "laws are obeyed not for their content, but because of the will of the instructor, *law* is not *advice* but *command*" [15] (pp. 153–154). If we insist on making law a content-dependent issue, particularly during a civil war, then we are expecting legal subjects to decide for themselves what is legally valid, and as a result, they "do not obey or they obey at their own discretion, i.e., they obey themselves not the commonwealth" [15] (p. 230). The entire legal system would be pointless or redundant.

The type of legal discourse favoured by Hobbes seems to rely on *pro ratione voluntas*, the characteristic motto of (at least) classical tyranny that, to put it mildly, is not quite a public relations success. However, what seems to be irrational at one level (say, the level of the balance of reasons any subject might have) may well be rational on a different level. Moreover, the fact that law claims to be content-independent does not entail that the content of a legal disposition cannot be rational. Perhaps we should say that law is arbitrary not in the sense of being capricious but in the sense that it could have been disposed otherwise. This is typical of what arbitrators or referees do, and in the face of substantive disagreement, we have reasons to abide by their decision instead of relying on our own assessment of the merits of what is to be done. Schmitt is then certainly right on the money as he claims, "The state is an empire of reason (this is a set phrase that first derives from Hobbes and not from Hegel), an *imperium rationis* (*De Cive* X.1), that transforms civil war into the peaceful coexistence of citizens" [6] (pp. 121–122). We should also bear in mind that as far as Schmitt is concerned "the essence of the state's sovereignty... must be juristically defined correctly, not as the monopoly to coerce or to rule, but as the monopoly to decide" [16] (p. 13).

In Schmitt's eyes, the state not only resolves political conflict, but it also provides, at the very least, protection in exchange for obedience: "No form of order, no reasonable legitimacy or legality can exist without protection and obedience. The *protego ergo obligo* is the *cogito ergo sum* of the state. A political theory which does not systematically become aware of this sentence remains an inadequate fragment. Hobbes designated this (at the end of his English edition of 1651, p. 396) as the true purpose of his *Leviathan*" [5] (p. 52). In his monograph on Hobbes, written in 1938, Schmitt holds that "The 'relation between protection and obedience' is the cardinal point of Hobbes's construction of state". The protection afforded by the state to individual subjects not only refers to other individuals, but it is also and mainly a way of

> confronting medieval pluralism, that is, power claimed by the churches and other "indirect" authorities, with the rational unity of an unequivocal, effective authority that can assure protection and a calculable, functioning legal system. To such a rational state, power belongs to the assumption of total political responsibility regarding danger and, in this sense, responsibility for protecting the subjects of the state. If protection ceases, the state too ceases, and every obligation to obey ceases. The individual then wins back his "natural" freedom. The "relation between protection and obedience" is the cardinal point of Hobbes' construction of state. It permits a very good reconciliation with the concepts and ideals of the bourgeois constitutional state. [17] (pp. 71–72)

It is quite revealing that Schmitt refers in 1938 to how the "legal state breaks on the pluralism of the indirect powers" [18] (p. 7) and speaks of Hobbes as "the great teacher

in the struggle against all types of indirect power" [18] (p. 131), the Nazi party being a typical indirect power that takes control of the state without assuming the responsibility of the political, without fulfilling the *cogito ergo sum* of the state, i.e. providing protection in exchange for obedience.

The absorption of the political by the state also allows for the separation between state and civil society, which in turn enables civil society to be autonomous or at least free from total politicization. Some people think that *everything* is political, even a game of bowling[7]. However, this would only make sense during a revolution, viz., when the political is no longer absorbed by the state but reabsorbed entirely by civil society. During revolutions, instead of being a neutral institution, the state becomes an instrument of a faction undertaking, say, a world proletarian revolution or a racial revolution. As a result, "Politics ceases to be a part of life, and becomes the whole of it. The philosophy which promises the end of politics, makes politics into the sole human end" [20] (p. 214).

It is sometimes assumed that Schmitt keeps democracy apart from liberalism, at least in the sense that liberal democracy is not a match made in heaven but mainly the result of "a highly contingent historical process" [21] (p. 3), because he preferred democracy over liberalism. However, Schmitt's point is mainly conceptual, and it might even go the other way, particularly if we keep in mind that "*The Concept of the Political* was yet another ingenious attempt to reassert the distinction between state and society" [22] (p. 33). According to Schmitt,

> The equation state = politics becomes erroneous and deceptive at exactly the moment when state and society penetrate each other. What had been up to that point affairs of state become thereby social matters, and vice versa, what had been purely social matters become affairs of state—*as must necessarily occur in a democratically organized unit*. Heretofore ostensibly neutral domains—religion, culture, education, the economy—then cease to be neutral in the sense that they do not pertain to the state and to politics. The total state appears as a polemical concept against such neutralizations and depoliticalizations of important domains, which potentially embraces every domain. This results in the identity of state and society. In such a state, therefore, everything is at least potentially political, and in referring to the state, it is no longer possible to assert for it a specifically political characteristic. [5] (p. 22, emphasis added)[8]

Thus, unadorned democracy is put on a par with revolution as it tends to politicize every aspect of life, which, in turn, necessarily puts into question the absorption of the political by the state. Clearly, this is not Schmitt's cup of tea. On the contrary, this is part of the problem caused by the era of revolution, a problem that underlies Schmitt's entire concept of the political. This is why Schmitt can be said to be Tocqueville's disciple, someone very much aware of the unrestrained march of democracy and worried about the toll this would take not only on liberalism but also on the state. Thus, although by "liberal" Schmitt often understands the denial of the political, Schmitt himself can then be said to be a liberal in the Hobbesian sense of the word, viz., a liberal aware of the close link between the inalienable rights of the individual and the state.

### 3. Rechtsstaat or Pluralism

In fact, were we to be too democratic or generous in our recognition of political agency, we would not be capable of preventing anti-democratic forces from taking over our own liberal, rule-of-law democracy that protects human rights, particularly the rights of minorities. This is the gist of Schmitt's argument against extreme pluralism, viz., the idea that the state is nothing but a political agent among others that must compete with the full plurality of social agents or corporations in total equipoise with them. In other words, the main problem with extreme pluralism is that "That is the way to lay the city flat/To bring the roof to the foundation/And bury all, which yet distinctly ranges/In heaps and piles of ruin"[9].

Pluralistic thought takes order as a given because it tends to arise in times when different groups seem to order themselves outside the framework of the sovereign state: "This seems to presuppose the priority of the state idea, for the state idea is, after all, nothing more than the idea of order. When order is lacking, ideas of the sovereign state achieve prominence, and political theorists become involved in the business of seeking order themselves. But when order is achieved, though ideas of the sovereign state may be undermined, the state idea, by definition, is not. It simply moves to the background" [23] (p. 159).

Now, "next to the primary political decisions and under the protection of the decision taken, numerous secondary concepts of the political emanate" [5] (p. 30). Once political unity is achieved, the primary concept of the political recedes into the background as the secondary concept of the political takes centre stage. If there is some kind of impartial authority in charge of resolving the conflict, then as far as Schmitt is concerned we should not talk about the political in this case, not in the primary sense of the concept anyway. The secondary concept of the political deals mainly with issues regarding public policy, not the political in the Schmittian and primary sense of the word.

In his rather critical review of the 1933 edition of *The Concept of the Political*, the National Bolshevik Ernst Niekisch wonderfully captures the gist of what goes on within the political unit in the Schmittian sense: "The friend–enemy distinction becomes senseless; it is no longer the political differentiation. Politics gets trivial; it becomes 'realization of law' or a similar dovish affair". This is due to the fact that

> Discrepancy is no "enemy"; it is an "opponent" with whom one sees eye to eye on the basis of a fundamentally equal level, with whom one can "duel" but only metaphorically. The will to existential annihilation is here displaced; weapons are thrown to the floor. There is mutual dialogue thanks to a feeling of a "deeper underlying commonality": there is a constant discussion. "The opponent is respected" as one's own kind; there is tolerance and "liberalism" anew. (...). Where there exists an unchallenged commonality about the final social, popular, and ideological basic questions, there is no "enemy" in the Schmittian sense. [24] (pp. 373, 375)[10]

Extreme pluralism is, thus, the worldview of those who deny the political since they believe "That by way of a rational discussion all conceivable oppositions and conflicts can be settled peaceably and justly, that one can speak about everything and allow oneself to speak with those like oneself". Thus, extreme pluralists claim that "public discussion", *without the intervention of authoritative institutions*, "finds the reasoned truth and the just norm. The discussion is the humane, peace-loving, progressive means, the opposite of every type of dictatorship and authority" [26] (p. 338). As far as extreme pluralism is concerned, the political "presents only differences of opinion, not a friend/enemy grouping" [26] (p. 345).

When it comes to Schmitt's conception of judicial reasoning, the political is totally off limits for judges. In his view, "the political conflict cannot be resolved in a judicial procedure". Actually, "as soon as the case is regulated by a conclusive, recognized norm, it does not simply lead to a genuine conflict. But if that type of regulation is not present, then procedure does not, in fact, take a judicial form. And a court, which in lieu of stable, preexisting, general norms decides a political conflict according to its own discretion, only appears to be a court" [26] (p. 389).

In fact, Schmitt does not expect judges to come close even to the secondary concept of the political because, in his eyes, judicial reasoning is politics-free. The judge is bound to the statute: "His activity is essentially normatively determined. He is not an independent representative of the political unity as such. Considered politically, this adjudication, which is entirely dependent on statute, is 'en quelque façon nulle'. (...). If democracy is basically political form, the judiciary, by contrast, is fundamentally nonpolitical, because it is dependent on the general statute". Even in a democratic state, adds Schmitt, "the judge must be independent if he is to be a judge and not a political instrument. The *independence*

of the judge, however, can never be anything other than the reverse side of his *dependence* on statutory law". The "purpose" of the independence of the judge, unlike that of a deputy of a legislative body, "is the *rejection* of the political. Everything that a judge as judge does is normatively determined and distinguishes itself from the existential character of the political, (…)" [26] (pp. 299–300). It would be difficult to find a more stringent defence of the autonomy of judicial reasoning vis-à-vis political reasoning.

This might come as a surprise, particularly in the light of the almost legendary debate between Carl Schmitt and Hans Kelsen on the nature of law and judicial reasoning. Indeed, in *Political Theology,* Schmitt seems to take exception to Kelsen's jurisprudence for being "concerned with ordinary day-to-day questions" and, therefore, for having "no interest in the concept of sovereignty" [16] (p. 12), which is a very accurate description of precisely what a judge should do according to Schmitt's own constitutional thought. Were we to stick to Schmitt's portrait in *Political Theology* of Kelsenian judicial reasoning, we would be hard-pressed to distinguish one from the other.

On the other hand, if we hear it from the horse's mouth, Kelsen's theory of adjudication appears to be a defence of judicial activism in comparison with Schmitt's: "Insight into the hierarchical structure of the legal systems shows that the contrast between making or creating the law and carrying out or applying the law by any means have the absolute character accorded to it by traditional legal theory, where the contrast plays such a role. Most legal acts are acts of both creation and law application. With each of these legal acts, a higher-level norm is applied and a lower-level norm is created" [27] (p. 70).

Kelsen's rather generous views on adjudication vis-à-vis Schmitt's rather old-fashioned account of judges as servants of the law go a considerable way towards explaining their debate on the guardian of the constitution and the legitimacy of constitutional review. However, their disagreement has been somewhat exaggerated. Although Schmitt is often taken to be the champion of an utter rejection of any kind of judicial review in the face of Kelsen's staunch defence of judicial involvement in constitutional reasoning, we would do well to recall that in his discussion of the separation of powers in *Constitutional Theory*—i.e., the institutional companion to the concept of the political—Schmitt holds that "I would like to affirm the competence of judicial review in regard to the constitutionality of simple statutes [*Gesetze*], for with no *influence* of the judiciary on the legislative branch the principle of the separation of powers still remains intact. The judiciary is generally not in the position 'to intervene and to influence' in the same manner as other state activities. It is bound to the law, and even if it resolves doubts about the validity of a law, it does not abandon the sphere of the purely normative. (…). Consequently, I would see no violation of the Rechtsstaat principle of the separation of powers in the judicial review of the validity of statutes [*Gesetze*], because there is no 'intervention' in the genuine sense" [26] (pp. 231–232).

Mind you, although Schmitt does not certainly want judges to go into the substance of the Constitution, the "statutes" referred to by Schmitt that may be subject to judicial review are no small potatoes since they cover constitutional laws in general. What remains off-bound for judges (and occasional parliamentary majorities) is what Schmitt calls the Constitution itself, e.g., the political decisions made by the constituent power like "the German Reich is a republic" or "state authority derives from the people". Schmitt's point is that "Everything regarding legality and the normative order inside the German Reich is valid only on the basis and only in the context of these decisions. They constitute the substance of the constitution" [26] (p. 78). In turn, according to Schmitt, "the practical meaning of the difference between constitution and constitutional law" is that, whereas "constitutional laws can be changed by way of Art. 76 [of the Weimar Constitution]", "the constitution as a whole cannot be changed in this way". For instance, "The German Reich cannot be transformed into an absolute monarchy or into a Soviet republic through a two-thirds majority decision of the Reichstag" [26] (p. 79).

At any rate, if we have been lucky enough to leave the primary concept of the political behind, or perhaps in the background, so that hopefully liberal democratic institutions, principles, and rules are in charge of resolving the ordinary agenda of political unity, this

means that our political unit by definition has achieved some kind of homogeneity. This is why exclusion is the price we pay for collective action. Political unity is the very contrary of a totally open house. Today, some people stand in favour of total inclusion or diversity. However, if taken at face value, total inclusion would entail that even National Socialists must be welcome.

To say this in Schmittian parlance, every time there is political unity an internal enemy has been declared[11]

> As long as the state is a political entity this requirement for internal peace compels it in critical situations to decide also upon the domestic enemy. Every state provides, therefore, some kind of formula for the declaration of an internal enemy. The πολέμῐος declaration in the public law of the Greek republics and the *hostis* declaration in Roman public law are but two examples. Whether the form is sharper or milder, explicit or implicit, whether ostracism, expulsion, proscription, or outlawry are provided for in special laws or in explicit or general descriptions, the aim is always the same, namely to declare an enemy. (. . .). More so than for other states, this is particularly valid for a constitutional state, despite all the constitutional ties to which the state is bound. In a constitutional state, as Lorenz von Stein says, the constitution is "the expression of the societal order, the existence of society itself. As soon as it is attacked the battle must then be waged outside the constitution and the law, hence decided by the power of weapons". [5] (pp. 46–47)

Thus, the very fact of setting up a liberal democratic constitution entails that the enemies of liberal democracy have been excluded. For some reason, however, liberals tend to pass over the exclusionary nature of political reasoning. This is why in 1928, during his inaugural lecture on Hugo Preuss and "his concept of the state and his position in German state theory" at the School of Business Administration in Berlin, Schmitt emphasized that "As impossible as it is for a state to be neutral on the question of its own existence, it is equally hard for a constitution to remain neutral on the political decisions which constitute its fundamental (positive) substance". Indeed, "The constitution would itself be brought to ruin, since it cannot, despite its neutrality, be neutral with respect to its founding principles" [28] (p. 364, n. 49).

However, not all liberals are blind to the inevitability of the political. In *Constitutional Theory*, Schmitt quotes approvingly the Italian liberal Mazzini: "'Freedom constitutes nothing', as Mazzini aptly stated". The Rechtsstaat contains basic rights and the division of powers but is not in itself a "state form. It is, rather, only a series of limitations and controls on the state, a system of guarantees of bourgeois freedom that makes state power relative. *The state itself*, which should be controlled, *is presupposed in this system*. The principles of bourgeois freedom could certainly modify and temper a state. Yet they cannot found a political form on their own". This is why "in every constitution with the Rechtsstaat component, there is a second part where principles of *political* form are bound up and mixed in with the Rechtsstaat one" [26] (p. 235).

This concern about the defence of the political order is not only at the centre of Schmitt's *The Concept of the Political* and his *Constitutional Theory*, but it also underlies Schmitt's first grand monograph—his study of dictatorship—and its emphasis in distinguishing between commissary dictatorship meant to restore the political order threatened by revolution and sovereign or revolutionary dictatorship meant to destroy the political order of Weimar:

> In practice [*in concreto*] the commissary dictatorship suspends the constitution in order to protect it—the very same one—in its concrete form. The argument has been repeated ever since—first and foremost by Abraham Lincoln: when the body of the constitution is under threat, it must be safeguarded through a temporary suspension of the constitution. Dictatorship protects a specific constitution against an attack that threatens to abolish this constitution. The methodological autonomy, as a legal problem, of the problem of law implementation becomes

most evident here. The dictator's actions should create a condition in which the law can be realised, because every legal norm presupposes a normal condition as a homogeneous medium in which it is valid. [29] (p. 118)

In 1931, Ernst Forsthoff, one of the most prominent disciples of Schmitt, under the *nom de plume* Friedrich Grüter, wrote a piece for the radical right-wing journal *Deutsches Volkstum* that on the basis of the concept of the political brings clearly to light the danger represented to the Weimar Republic by National Socialism, particularly its strategy of running for office as legally as possible (this is all the more remarkable since Forsthoff himself ended up supporting National Socialism):

The liberal Rechtsstaat must remember that it is a *state*, i.e., an essentially political power association [*Machtverband*] and not an unpolitical legal association, it must cruelly burst the bubbles of the old liberal hopes and restrict liberties, partly revoke them altogether, in order not to go under on their account. The liberal Rechtsstaat dies of the legality of their opponents. In this age, we find ourselves in this process of the self-abolition of the liberal Rechtsstaat of liberties. The state, which has been irrevocably banned with gusto in the apolitical-ethical sphere of law, must obtain its adequate position in the realm of the political, so that it will be forced to make a friend–enemy distinction, through which Carl Schmitt sees determined the essence of the political. There is scarcely a more impressive confirmation of the Schmittian definition of the political than this indicated mechanism[12].

In *Legality and Legitimacy*, written in July 1932, Schmitt fears once again that pluralism or "value neutrality" entails that "Any goal, however revolutionary or reactionary, disruptive, hostile to the state or to Germany, or even godless, is permitted and may not be robbed of the chance to be obtained via legal means". Schmitt then quotes back his treatise on *The Guardian of the Constitution* (1931):

This dominant interpretation of Article 76 [on constitutional amendment] deprives the Weimar Constitution of its political substance and its foundation, making it into a neutral amendment procedure that is indifferent toward every content. More importantly, this amendment procedure is even neutral toward the currently existing state form. Under these circumstances, justice requires that all parties be given the unconditional equal chance to produce the majorities that are necessary, with the help of the valid procedure for changing the constitution, to achieve their desired goal—soviet republic, national socialist Reich, economic-democracy state of labor unions, corporativist state organized by professions, monarchy of the traditional form, some type of aristocracy—and bring about another constitution. [32] (pp. 48–49)

We must be very wary then of "Legality and the Equal Chance for Achieving Political Power", "The method of will formation through simple majority vote is sensible and acceptable when an essential similarity among the entire people can be assumed. For in this case, there is no voting down of the minority. Rather, the vote should only permit a latent and presupposed agreement and consensus to become evident". Even liberal democracy, then, "rests on the presupposition of the indivisibly similar, entire, unified people, for them there is, then, in fact and in essence, no minority and still less a number of firm, permanent minorities. The process of determining the majority comes into play, not because finding what is true and proper has been renounced on account of relativism or agnosticism—that would be, in view of the momentous political decisions at issue here, a suicidal renunciation". Schmitt adds that Hans Kelsen himself "concedes that this would be possible 'only in relatively peaceful times', thus only when it does not matter". At the end of the day, liberal democracy itself assumes that

by virtue of being a part of the same people, all those similarly situated would, in essence, will the same thing. If the assumption of an indivisible national

commonality is no longer tenable, then the abstract, empty functionalism of pure mathematical majority determinations is the opposite of neutrality and objectivity. It is only the quantitatively larger or smaller, forced subordination of the defeated and, therefore, suppressed minority. The democratic identity of governing and governed, those commanding and those obeying, stops. The majority commands and the minority must obey. Arithmetic calculability also stops because one can only reasonably produce a sum from that which is very similar. [32] (pp. 27–28)

We trust majorities because we assume a substantive principle of justice, e.g., that they will not establish themselves as the permanent legal power. Liberal democracy is not simply then about mathematics or majorities. We do want our own party to win, but first of all, we are partisans of liberal democracy. This comes neatly to the surface in the following portrait by Schmitt of the British parliament in 1832, but it may be easily generalized to refer to any kind of parliament: "The distinction between Tories and Whigs generalizes itself into that between conservatives and liberals, which initially does not signify genuine class opposition. Instead, this distinction is rendered relative by the independent, enclosed unity of the nation that encompasses both parties, and, as such, it presents only differences of opinion, not a friend/enemy grouping" [26] (p. 345).

In a preliminary remark to the published version of her dissertation, *The Concept of a Political Party in the System of Political Liberalism*, Johanna Kendziora—her thesis was the best one Schmitt supervised during his first sojourn in Berlin—states that (not without clearing up that the manuscript was finished on 6 January 1933): "In the meantime, the political reality in Germany has drawn consequences from the theses on political parties and the party expounded in this work"[13]. One of the theses defended in her dissertation was that: "It is a consequence of the political that every system declares as 'inimical to the state' any actions that endanger its existence, and that it can only recognize the objectives that agree to coexist with the conservation of the specific type of state" [31] (p. 254).

Karl Loewenstein, a Jewish jurist and political scientist who considered himself to be the nemesis of Carl Schmitt—he had been his colleague in Weimar, fled to America after the Nazi takeover in 1933, and returned to Germany as part of the allied team in charge of bringing former Nazi officials to trial—in 1937 published a paper about "Militant Democracy and Fundamental Rights", which comes down to being an homage to Schmitt's concept of the political, particularly about the need of a declaration of the internal enemy: "[liberal] democracy is at war, although an underground war on the inner front. Constitutional scruples can no longer restrain from restrictions on democratic fundamentals, for the sake of ultimately preserving these very fundamentals. The liberal-democratic order reckons with normal times" [34] (p. 432).

Indeed,

democracy and democratic tolerance have been used for their own destruction. Under cover of fundamental rights and the rule of law, the anti-democratic machine could be built up and set in motion legally. Calculating adroitly that democracy could not, without self-abnegation, deny to any body of public opinion the full use of the free institutions of speech, press, assembly, and parliamentary participation, fascist exponents systematically discredit the democratic order and make it unworkable by paralyzing its functions until chaos reigns. They exploit the tolerant confidence of democratic ideology that, in the long run, truth is stronger than falsehood and that the spirit asserts itself against force. Democracy was unable to forbid the enemies of its very existence the use of democratic instrumentalities. Until very recently, democratic fundamentalism and legalistic blindness were unwilling to realize that the mechanism of democracy is the Trojan horse by which the enemy enters the city. To fascism in the guise of a legally recognized political party were accorded all the opportunities of democratic institutions. (. . .). Democracies are legally bound to allow the emergence and rise of anti-parliamentarian and anti-democratic parties under the condition that they

conform outwardly to the principles of legality and free play of public opinion. It is the exaggerated formalism of the rule of law, which, under the enchantment of formal equality, does not see fit to exclude from the game parties that deny the very existence of its rules. [34] (pp. 423–424)

As if he was foretelling what Schmitt would say after the Second World War on his own admonitions about the dangers visited upon those regimes that are not willing to make a stand against their internal enemies, Loewenstein warns that "the lack of militancy of the Weimar Republic against subversive movements, even though clearly recognized as such, stands out in the post-war predicament of democracy both as an illustration and as a warning" [34] (p. 426).

At the end of Section 4 of *The Concept of the Political* on the dangers to the state represented by extreme internal pluralism, Schmitt holds that "we shall attempt to show below (Section 6)" that "the concept of the political yields pluralistic consequences, but not in the sense that, within one and the same political entity, instead of the decisive friend-and-enemy grouping, a pluralism could take place without destroying the entity and the political itself" [5] (p. 45). In other words, according to Schmitt, the rejection of internal pluralism goes hand in hand with the endorsement of external pluralism.

Schmitt makes good on his promise at the very beginning of Section 6: "every theory of state is pluralistic, even though in a different way from the domestic theory of pluralism discussed in Section 4. The political entity cannot by its very nature be universal in the sense of embracing all of humanity and the entire world". According to Schmitt, "The political entity presupposes the real existence of an enemy and therefore coexistence with another political entity. As long as a state exists, there will thus always be in the world more than just one state. A world state which embraces the entire globe and all of humanity cannot exist. The political world is a pluriverse, not a universe" [5] (p. 53).

The reason for this is that in Schmitt's view, Humanity as such "has no enemy, at least not on this planet. The concept of humanity excludes the concept of the enemy, because the enemy does not cease to be a human being—and hence there is no specific differentiation in that concept". The fact that

wars are waged in the name of humanity is not a contradiction of this simple truth; quite the contrary, it has an especially intensive political meaning. When a state fights its political enemy in the name of humanity, it is not a war for the sake of humanity but a war wherein a particular state seeks to usurp a universal concept against its military opponent. At the expense of its opponent, it tries to identify itself with humanity in the same way as one can misuse peace, justice, progress, and civilization in order to claim these as one's own and to deny the same to the enemy. (...). Here, one is reminded of a somewhat modified expression of Proudhon's: whoever invokes humanity wants to cheat. To confiscate the word humanity, to invoke and monopolize such a term probably has certain incalculable effects, such as denying the enemy the quality of being human and declaring him to be an outlaw of humanity, and war can thereby be driven to the most extreme inhumanity. [5] (p. 54)

Thus, the political use of moral terms like Humanity is particularly dangerous. Although it is a moral concept meant to be all-inclusive (at least as far as human beings are concerned), once employed in the political fray, it becomes an asymmetrical "counterconcept", viz. a concept that is "unequally antithetical" so that "one's own position is readily defined by criteria which make it possible for the resulting counterposition to be only negated" [35] (p. 159). Thus, the attempt to exclude all exclusion ends up bringing about the worst of exclusions.

Perhaps the clearest explanation given by Schmitt on "the especially intensive political meaning" of the term "Humanity" comes up in his *Glossarium*:

Your Humanity: a simple syllogism. Every human being is extraordinarily likable to me... Every one! Without exceptions! What a Humanity! What does it follow

from this? Think it over for a moment. Tax your wits. It follows from this something as easy as: someone who is not extraordinarily likable to me is not human. If he is inhuman or subhuman is another question. This is quite a totally perfect, humanitarian syllogism: Every human being is good and noble: Carl Schmitt is not good and noble; therefore, Carl Schmitt is not human. This, gentlemen, is the quintessence of today's humanism; nothing more. Hence, it is about discrimination and criminalization. [36] (p. 254)

Schmitt's explanation about the use of humanity as a political weapon can be illustrated by the vote of Foussedoire during the trial of Louis XVI as stated by Victor Hugo's *Ninety-Three*: "I have a horror of human bloodshed, but the blood of a king is not human blood" [37] (p. 225). This also explains the very hard and finally unsuccessful time had by Raymond Desèze, the criminal lawyer in charge of pleading on behalf of Louis XVI, as he argued during the trial that "Louis is nothing but a man, an accused man" [38] (p. 61).

In fact, the French Revolution itself, "despite its ideas of humanity and general brotherhood of all peoples, presupposes the French *nation* as a historically given entity", thereby proving that inclusion by exclusion is unavoidable. "Different elements can contribute to the nation and to the consciousness of this unity, such as common language, common historical destiny, traditions and remembrances, and common political goals and hopes. What is definitive is the commonality of historical life, conscious willing of this commonality, great events and goals" [26] (pp. 261–262).

Perhaps the most convincing defence of the inevitability of exclusion in political reasoning also comes up in Schmitt's *Glossarium*: "The fate of my little work 'The Concept of the Political' is already a component of this concept itself. This is the highest that can be said to sustain its glory; it is the safest symptom of its existential authenticity. The essay catches sight of the criterium of the political in the distinction between friend and enemy, and behold: this essay arouses friendship and enmity and proves its energy of grouping friend and enemy whatever it meets" [36] (p. 106). Those who deny the concept of the political, the conceptual distinction between friend and enemy as constitutive of the political, cannot avoid becoming *eo ipso* enemies of *The Concept of the Political*, thereby giving Schmitt the upper hand.

At the end of the day, Schmitt does not quite recommend enmity or exclusion, for that matter, for the simple reason that they are unescapable: "the political is unavoidable" [26] (pp. 206–207)[14]. Lord Rabbi Jonathan Sacks strikes quite a Schmittian chord as he explains that "Groups unite and divide. They divide as they unite. Every group involves the coming together of multiple individuals to form a collective Us. But every Us is defined against a Them, the ones not like us. The one without the other is impossible. Inclusion and exclusion go hand in hand" [40] (p. 30)[15] Schmitt claims that not even a liberal democratic state, "let us say the United States of America", is close to "allowing foreigners to share in its power or its wealth. Until now there has never been a democracy that did not recognize the concept 'foreign' and that could have realized the equality of all men" [42] (p. 11)[16].

Section 7 of *The Concept of the Political* lays the foundation stone upon which Schmitt builds his exclusionary church. Although Schmitt starts off by saying that "One could test all theories of state and political ideas according to their anthropology and thereby classify these as to whether they consciously or unconsciously presuppose man to be by nature evil or by nature good" [5] (p. 58), we may well rephrase this discussion in terms of the opposition between what we might call anthropological realism and idealism, or between the tragical and the utopian vision on human nature. Once the utopian vision gets the better of the tragical one, perhaps with the best of intentions, "the effort to figure out what is going on has been hijacked by an effort to legislate the correct answer" [45] (p. 294). Anthropological realism does not entail that life is necessarily a valley of tears, so there is nothing we can do about it, or that cooperation is impossible. On the contrary, realism is the only responsible way to ameliorate reality and realism shows that cooperation is always somewhat parochial. Precisely, if we want to cooperate, we must form groups. "We need groups, we love groups, and we develop our virtues in groups, even though those

groups necessarily exclude nonmembers. If you destroy all groups and dissolve all internal structure, you destroy your moral capital" [46] (p. 358).

The exclusionary nature of political reasoning, supported by cognitive psychology, cuts across political ideologies, and rightly so, since it is a constitutive feature of political reality that we can only ignore at our own peril. A liberal like David Miller is perfectly aware of the fact that political units "are likely to function most effectively when they embrace just a single national community. The arguments here all appeal to the political consequences of solidarity and cultural homogeneity. They focus on the important role played by trust in a viable political community" [44] (p. 90). In a similar sense, a social democrat like Yael Tamir, for her part, holds that "One cannot create communities that are both meaningful and entirely open: the more meaningful a community is to its members the more exclusive it would be to all others". This is why "inclusion, not exclusion, has its costs" [47] (p. 157).

Tamir also explains that "Sharing resources and social responsibilities is logical for those who anticipate spending a lifetime altogether; it is far less so for those who expect to move from one political entity to another". This is why "democratic welfare states are grounded in closure that ensures the persistence of stable and continuous communities, allowing for lifelong and often cross-generational bonds to develop. In a world of permeable borders, where social stability is no longer guaranteed, the 'haves' have less reason to share and the 'have-nots' are left unprotected". The irony is that since "the close affinity between welfare policies and political closure was often ignored", "consequently, both progressives and conservatives are surprised to find out that while they preach for the one, they may end up with the other" [47] (pp. 98, 176).

Schmitt realizes that the realism of the concept of the political "can frighten men in need of security. Without wanting to decide the question of the nature of man one may say in general that as long as man is well off or willing to put up with things, he prefers the illusion of an undisturbed calm and does not endure pessimists. The political adversaries of a clear political theory will, therefore, easily refute political phenomena and truths in the name of some autonomous discipline as amoral, uneconomical, unscientific and above all declare this—and this is politically relevant—a devilry worthy of being combated" [5] (pp. 65–66).

To be sure, Schmitt, at times, seems to give the impression that he bemoans the loss of the political: "How appalling a world is in which there is no more an exterior and only an interior" [36] (p. 28). This seems to be the scenario described by the character Mustapha Mond in Aldous Huxley's *A Brave New World*: "Where there are wars, where there are divided allegiances, where there are temptations to be resisted, objects of love to be fought for or defended—there, obviously, nobility and heroism have some sense. But there aren't any wars nowadays. The greatest care is taken to prevent you from loving anyone too much. There's no such thing as a divided allegiance; you're so conditioned that you can't help doing what you ought to do". "And if ever, by some unlucky chance, anything unpleasant should somehow happen", "there's always some *soma* to calm your anger, to reconcile you to your enemies, to make you patient and long-suffering" [48] (pp. 209–210).

Nevertheless, as we have seen, Schmitt claims that the political is unavoidable, it is a reality that cannot be wished or idealized away. That is the whole point of the concept of the political. This is also why "Nowhere in his writings can one detect a desire on his part to perpetuate crises as a means of escaping the tediousness of everyday bourgeois existence" [49] (p. 55). The very title of the essay that accompanies "The Concept of the Political" since 1932, "The Age of Neutralizations and Depoliticizations", is somewhat ironical, since in that essay Schmitt shows that every depoliticization turns into a new politicization: "It would therefore be best, means Schmitt, to leave politics there where it was at the beginning of modernity, viz. in the religiously neutralized, sovereign state" [50] (p. 127). What Schmitt truly bemoans then is the golden age of the state because, during that time, the *jus publicum europaeum* was at the peak of its game in keeping the political at bay. The problem is that in the meantime, revolution broke out, and in spite of its promises

to get rid of it, the political remains. Paraphrasing Cicero's dictum: *politicum manet*, *res publica cessat*[17].

## 4. Antisemitic or Jewish-Friendly

There is no denying that during most of his life, Schmitt held antisemitic views. Actually, in his memoirs, Edgar Feuchtwanger, son of Ludwig Feuchtwanger—the Jewish liberal CEO of Duncker & Humblot and personal editor of Schmitt's works for that publishing company until 1932—holds that "once the Nazis came to power, Schmitt dropped my father 'like a hot potato', just like he did with other Jewish friends" [51] (p. 54)[18].

However, Edgar Feuchtwanger himself explains that:

Schmitt was a favourite author of my father, even his friend, as proven by the published voluminous correspondence. Through his works, that appeared under the care of my father, Schmitt became the most prominent intellectual critic of liberal parliamentarian democracy. In one of his most influential publications, Schmitt argued that the parliamentary supremacy of the liberal age was based upon the idea that the right policy is shaped through debates and dialogue. But this process could only be carried by the elites and became inoperable through the irruption of the masses in politics. Even my father, who was quite a liberal and tolerant man, shared this fear of the mass that was totally spread in the German intelligentsia. (...). Even a basically liberal man like my father must have realized in the 1920s on what a thin ice the neutral Weimar state was built, so that Schmitt's sharp formulations bringing out the state of exception did not surprise him [51] (pp. 53, 55).

Edgar Feuchtwanger himself also holds that Schmitt "*changed sides* a few months after Hitler came to power" [51] (p. 54, emphasis added). To be sure, Feuchtwanger's remark may be taken to refer to the change in Schmitt's political stance vis-à-vis National Socialism, not to Schmitt's antisemitism, but only because Schmitt may well have kept his antisemitism to himself (e.g., in his personal diaries) so that he managed to prevent his loyal editor from being abreast of it. However, that is precisely the point: Schmitt pulled this off because his antisemitism did not affect his published work for Duncker & Humblot, including what Raphael Gross calls Schmitt's "best-known book from the Weimar Republic period, *The Concept of the Political*". If Gross were right in saying that "[Schmitt's] concept of the enemy", as formulated in his essay on the political, "is at least grounded in the same intellectual structure as his antisemitism" [52] (p. 232)[19], Schmitt would have hardly been a favourite author of Ludwig Feuchtwanger, the CEO of Duncker & Humblot at the time. Antisemitic as he was during most of his life, this fact did not affect Schmitt's concept of the political, as also shown in the previous two sections.

In his later age, Schmitt claimed that: "I cannot be denazified, because I cannot be nazified" [33] (p. 507). In fact, after one of his students visited him in Berlin after curfew in April 1945, the occupying Russian authorities asked Schmitt to corroborate the story. Schmitt, revealingly, assumed he was being questioned about his National Socialist past and immediately launched into a major defence [54] (p. 146). This is why in front of the Russian commission, Schmitt compared his own collaboration with Nazism with an experiment made by Max Pettenkofer, a German natural scientist who, at the beginning of the XX century, argued that infectious diseases were not caused by a bacillus alone, but the individual's predisposition toward disease was paramount. In order to prove his point, he drank a glass of water contaminated with cholera bacilli in front of his students and managed to stay healthy. Obviously, Schmitt's conclusion was, "You see, I did exactly the same thing. I drank the Nazi bacillus, but was not infected" [54] (p. 148). Schmitt was finally released by the Russians. It goes without saying that Schmitt's explanation does not make any sense about himself, but it certainly applies to his concept of the political at the very least. At any rate, we should beware of mistaking support of the autonomy of the political for National Socialism (or antisemitism, for that matter). Otherwise, only those who reject the concept of the political and its attendant exclusionary nature by claiming that they have no enemies whatsoever would be off the hook of being charged with holding National Socialist views. Moreover, those who take Schmitt gravely to task for having



embraced National Socialism surely do so because they consider themselves to be *enemies* of National Socialism. In other words, even those who reject the concept of the political and its emphasis on the exclusion would surely make an exception for the exception regarding, say, National Socialism.

Speaking of the devil, there is no beating about the bush with the fact that for all his rejection of revolution, Schmitt ended up collaborating with the Nazi regime. However, this cannot be chalked up to the concept of the political. Schmitt's decision to go along with National Socialism went against everything his concept of the political stood for. This was clear from the beginning to, for instance, Waldemar Gurian, "perhaps the most devoted early disciple of Schmitt" [55] (p. 51)[20]. Of Jewish origin himself, Gurian wrote many articles from exile in order to expose the deep transformation experienced by Schmitt's ideas and allegiances. Gurian reminded his readers that Schmitt began his career with the assistance of Jewish liberals such as Moritz Julius Bonn, "he had Jewish friends and admired Jewish scholars such as Hugo Preuss", "he had been a major exponent of political Catholicism, and still was a Roman Catholic. More damaging was the public disclosure of Schmitt's affiliation with the presidential system and his former opposition to National Socialism, which he had once called 'organized mass insanity'". In fact, Gurian taunted the Nazis for their credulity since "Schmitt could never be a true National Socialist" [55] (pp. 224–225). There is then reason to believe that "Schmitt had never paved the way for the Nazis, shared their ideological beliefs, nor provided the legal foundations for the Nazi state, as often alleged" [50] (p. 282). Gurian's charges were taken up almost verbatim in 1936 by the SS publication *Das Schwarze Korps*, who accused Schmitt of being "not a National Socialist, but a Catholic Thinker and an opportunist with numerous Jewish connections". In fact, the SS inferred that the disgraceful conference organized by Schmitt in 1936 on the Jewish influence in German legal thought "was an attempt by Schmitt to rehabilitate himself within National Socialism" [55] (p. 234).

In fact, a very brief recap of the structure of the concept of the political—including the 1933 edition of the book—would be enough to show that there is nothing National Socialist about it. First of all, whereas Schmitt's concept of the political aims at keeping conflict to a minimum, the very point of National Socialism was to criminalize and to do away with all its enemies—in the face of the normative constraints of the political—and actually with the very idea of the political: "Someone who does not contemplate the world but wants to change it must find it wanting. It gives me the creeps this pathos of moral indignation that arises when someone wants to create the world anew in order to let off their will to power and annihilate their enemies" [36] (p. 33).

Moreover, Nazi jurists took Schmitt's distinction between friend and enemy to be too liberal, too Hobbesian, in that it was disagreement and conflict that explained the need for a state[21]. Furthermore, as with every other type of revolutionary discourse, National Socialists criminalized their every enemy. National Socialism also embodied the idea of internal pluralism since it had no real use for the notion of the state as a neutral instance capable of resolving political conflicts; within Nazi Germany, there was only one option available, and that was that of the Nazi party. Moreover, "Even in Hobbesian terms, the National Socialist state is no sovereign state but a pervertedly powerful form of the state of nature, where *no one* is sure if he or she is friend or enemy to fellow citizens or to the regime, constituted as it is by an irresponsibly destructive, particularist group of fanatics" [56] (p. 277).

There is no place, either, for external pluralism in National Socialism since its foreign policy was neatly imperialistic. As for its anthropology, National Socialism was a political religion that deified human beings—or some of them, anyway. As far as Hitler was concerned, "the Aryan was the 'Prometheus of mankind', who, in the revolt against the gods, had acquired divine attributes that made him the prototype of genius and creativity" [57] (p. 60)[22]. We should also bear in mind that, on the contrary, in Schmitt's eyes, the state nourishes itself "from Prometheus' entrails" [59] (p. 203). In the light of all of these considerations, National Socialism may well have been used by Schmitt himself

in the last section of his book on the concept of the political as the paragon itself of the negation of the political.

Moreover, if Schmitt's concept of the political is often mistakenly taken to task for being the embodiment of antisemitism (or worse), a case may well be made that Schmitt's concept of the political may be interpreted as a Jewish-friendly text. George Schwab points out that "On more than one occasion, Schmitt would tell me that Jews understood his thoughts better than anyone else" [60] (p. 164). It is well known that Schwab is the Christopher Columbus of Anglo-Saxon scholarship on Schmitt due to his translations of and scholarship on the author of *The Concept of the Political*, and that he is Schmitt's English-language executor. It is perhaps less well known that Schwab is a Jewish Holocaust survivor, during his youth was a militant Zionist activist, and as he met Schmitt in Plettenberg in the late spring of 1957, Schmitt's former assistant in Berlin 1934-1935, Bernhard von Mutius, suspected that Schwab was at least a "half-Jew" and tried to warn Schmitt about it in writing (18 September 1957)[23], but to no avail. Already in October 1957, Schmitt, oddly enough, would refer to Schwab in his *Glossarium* as his "friend" [36] (p. 366), and they would remain close friends. This goes some way towards explaining why Schmitt also had blown his own horn as he wrote to Ludwig Feuchtwanger in 1931 as the second edition of the book was underway, "It is from Zionists that I have received the best statements of approval to *The Concept of the Political*" [61] (p. 353)[24].

Political Zionism naturally entailed "a true revolution addressed against the historical destiny of Jews" since "it invites them to get rid of a terrible weight: the mistrust with regard to the political" [63] (p. 119). The revolutionary nature of Zionism for Jews was particularly felt among German Jews, who were totally assimilated in Germany. This explains why if, by the beginning of the XX century, they were c. 1% of the German population, Zionists represented 1.2% of all German Jews. Actually, "Most [German Zionists] were said to be *Ostjuden*, recently arrived from Eastern Europe" [64] (p. 289).

To give just a taste of the way in which some German Zionists were taken by Schmitt's work, we may consider a couple of examples. Let us see first the case of Fritz Bernstein, to whom Schmitt refers in a letter to Carl Muth of 23 December 1927 as "the author of 'Antisemitism as a Group Phenomenon'" and one of the "Zionists who have answered in a specially vivid way to my essay 'The Concept of the Political'" [65] (p. 147)[25].

Bernstein, who eventually changed his first name to Peretz in Israel, was born in 1890 in Meiningen, Germany. He received only intermediate education in trade and commerce. Before his military service, Bernstein went to Rotterdam for an apprenticeship, and after military service in Germany, he returned to Holland in 1909 to work at a coffee trade firm in Rotterdam. Soon after, he became the son-of-law of the Jewish owner. Eventually, he started his own firm.

Fritz Bernstein became quite active in the Dutch Zionist movement and was president of the Dutch Zionist Federation for the period 1930–1934. In 1936, he emigrated to Palestine and became a member of the non-socialist and non-religious General Zionist Party. In 1948, he was one of the 36 signatories of the Declaration of Independence and was elected as part of the first composition of the Knesset. He was also minister of economic affairs in two cabinets and a member of Parliament from 1949 until 1965[26].

Bernstein's book was written in German in 1923, but it was not easy for him to find a publisher. He submitted the manuscript to the Jüdischer Verlag in Berlin, which accepted the book and published it in 1926. The book did not quite receive much attention at the time due to the fact that the author did not have a university degree. Furthermore, he had lived outside Germany for about 16 years and did not belong to any German academic networks and, in the words of Bernard M. S. van Praag, "The title suggested that the main subject was anti-Semitism and since most German university professors at that time were not very Jew-friendly, to put it mildly, they were not interested in what a non-doctored Jewish businessman from Holland, publishing at an outspoken Jewish publishing house, could have to say about a subject that could only be interesting to those in Jewish/Zionist circles". On top of this, "the book itself was not written in the usual academic style of the

day. It did not contain the typically German half-page long sentences, it did not quote other authors, it did not contain the usual irrelevant footnotes and finally there were no references at all". In short, "in the 1920s it was far ahead of its time stylistically" [66] (p. xi). Its English translation—which bears the more appropriate title: *The Social Roots of Discrimination: The Case of the Jews*—appeared in 1951. It remains to this very day an insightful analysis of antisemitism that can compete with any later study in the field.

Although Bernstein was interested in explaining antisemitism, out of his research "there grew a general group theory to which antisemitism was related only as a very small, though specific, aspect of a general phenomenon". According to Bernstein, "all groups are exclusive; they are closed to outsiders, let the admission of strangers depend on the fulfilment of various more or less difficult conditions, and even where their expansive tendencies are most pronounced, they regard new members with diffidence and do not admit them to the full enjoyment of membership rights for a long time". Some groups may be better equipped for absorption of new elements, particularly "when sufficiently conscious of their own strength", provided new members "are willing to assume the new characteristics and divest themselves of the old ones". But the fact remains that as far as groups are concerned, "Inner homogeneity is maintained by exclusivity". Even "the most noble, charitable and honestly altruistic of men creates a boundary between humanity as a whole and an—always comparatively small—number of individuals who within the sphere of this friendly feelings enjoy preferential treatment" [66] (pp. 7, 164–165, 111–112).

Group formation, in turn, goes hand in hand not only with exclusivity but enmity: "*whenever groups are formed, the most remarkable characteristic is some latent or apparent conflict*: even the most harmless of clubs is essentially a closed unit, closed, that is to say, towards the outer world". Unfortunately, "the liberal will not be overly inclined to make a close investigation of enmity relations, as he is *a priori* convinced that they ought to fade away before his ethical postulates" [66] (pp. 108–109, 29).

Not even the invocation to humanity is an exception to the close association between group formation and enmity. A group formed for the sake of humanity will naturally tend to remove any obstacle to the general benefit of humanity, so much so that

> the idea of this mission not only justifies defensive self-preservation, but most vigorous deployment and even a powerful struggle for expansion. The war is not only just, but holy. To reach its aim, no sacrifice is too great, particularly if those to be sacrificed are those who, actually or allegedly, stand in the way of its attainment; hostile action which otherwise would meet with general condemnation becomes a sacred charge, for the group's mission ordains a relentless struggle against all those who resist the benefit of the mission. In this way, the group develops its group ideology into an idea of justification, which supplies it with the psychological prerequisites for the expression of enmity to any desired extent. The power of the mission idea and the sublimity of the mission are commensurate with the depth of the feeling of superiority. [66] (pp. 139–140)

Hence, even in the case of a group who acts on behalf of humanity, to the extent that they form a group, "admission to the group is only possible on the group's own conditions; and to members the outsider is always, more or less consciously, a potential enemy, or at least someone who, in respect of those who belong to the group, lacks some valuable and important quality". The fact is that "group formation as such never ceases", even among those "prophets and moralists" who "have ever found their ideal of humanity in a brotherly community of all men, where all separating influences so obviously connected with the existence of the uncounted group combinations according to race, nation, tribe, language, religion, caste, trade, class, and so forth, will have disappeared". Little wonder, then, that "the ideal of a brotherhood of all men has never been realised or even approximated". In fact, the very purpose of achieving the rather modest ideal of living together peacefully "have been doomed to failure, and have even themselves become abundant sources of enmity and strife" [66] (pp. 107–109, 103)[27].

Now Bernstein's thesis is that "Antisemitism appears as a special form of that group enmity which directs itself against ethnical minority groups of inferior strength" and argues that "Only because the Jews live everywhere as dispersed, weak and defenseless minority groups, *does that enmity which exists between groups everywhere*, assume in respect of the Jews such a particularly dangerous, deleterious, destructive character". This "tragedy of the Jewish minority groups" is "only excelled by their own blindness, which prevents them from the full realisation of the true nature of their situation" [66] (pp. 287, 291 [emphasis added], 261).

Indeed, the "sufferings of a certain group may have stultified its sensitivity, impaired its consciousness of value, *paralysed its capacity for acting to such an extent that it is no longer capable of any vigorous expression of enmity*". Hence, the stimulus "to be awakened from the stupor of slavery" must be "inculcated from outside". Bernstein revealingly adds that "The renascence movement of an ethnical-national group, for instance that of the Jews, follows the same line". Thus, the problem with permanent persecution is not only that it makes "heavy demands upon the power of resistance of the group", but that it "may finally undermine its will to exist. And for this mental collapse armistice is, as always, more dangerous than open war. At the very moment when relations have somewhat improved for the time being, terror of renewed persecution makes itself master of the persecuted minority group; only then is it overcome by the full misery of mental servitude; it loses the will to exist independently, and its members desire to be absorbed within the majority group" [66] (pp. 141 [emphasis added], 224–225).

Bernstein seems to have the situation of German Jewry particularly in mind as he remarks that the Jews who, after the First World War, "expected miracles from solemn declarations, political emancipation, changes of government or special protective legislation, experienced bitter disappointments; the legal privileges were in part granted, but that did not mean the end of the enmity against the Jews" [66] (p. 255). In fact, in 1923, Bernstein made the rather prophetic claim that:

> We shall again be shocked, we shall again cry out in despair, when tomorrow again Jews are, somewhere in the world, murdered, tortured, outlawed; we shall appeal to the conscience of the nations and call our prosecutors to account for their deeds, even as we are prepared to account and be responsible for our every action. But we should not blind ourselves to the realisation that no penitential sermons can change human nature, that no indignation can prevent enmity from transforming itself into hostiles desires, that *the phenomenon of group enmity cannot be banished from the earth by exhortations*, and what whatever has been done to bring the world to a more peaceful condition has so been done by measures calculated to affect human nature as it is and not as it should be. [66] (p. 290, emphasis added)

It is no wonder then that Bernstein took to Zionism[28], being perfectly aware that "A Jewish nation which lives in close settlement within its own country will probably be exposed to the hostility of the surrounding nations, and live in alternating states of war and peace, as has ever been the way of the world". However, adds Bernstein, "*the enmity between the Jews and their neighbours will then be no more than a normal enmity between one nation and the other*, and not that onesided and accursed hatred which has haunted the fragments of a tortured people through twenty centuries and over the whole of the inhabited world" [66] (pp. 291–292, emphasis added). To rephrase Bernstein's theory in Schmittian parlance, antisemitism is a form of political reasoning gone wrong, a form of inclusion by exclusion that ends up discriminating against those excluded, the enemies.

The *Jüdische Rundschau* [Jewish Panorama], the organ of German Zionism, also seems to be a fertile place to illustrate Schmitt's influence on Zionism. Let us take, for instance, an article about "The Jewish Question and Democracy" published by Arthur Prinz—at the time professor of economics at Humboldt University in Berlin—in March 1932. Much of Prinz's research and writing focused on the works of Marx as well as the relationship between German Jews and the economy. Just to give an inkling of the political taste of this

newspaper, at the time of the Reichstag elections of May 1924, it recommended a vote for the SPD, the only party faithful to "the great ideal claims of equal rights and liberating humanity" [67] (p. 244).

Prinz starts out by saying, "For most German Jews the snowball of antisemitism not only brings with it a serious threat to their existence, but also a deep shock to their political image of their worldview. For, under the democratic constitution drafted by Hugo Preuß and received with enthusiasm by the major part of German Jewry, a *Jew-hatred* has arisen that Germany has not known for generations". Prinz grants that under the *Kaiserreich* Jews had also experienced social and governmental antisemitism that kept them apart from the public and official spheres, but at any rate at that time "we lived in security and growing welfare and expected the best from the victorious energy of liberal and democratic thought". Prinz adds perhaps unknowingly in the footsteps of Bernstein, that "Nowadays, however, the Weimar Constitution has been in force for twelve years", "Jews haver risen to the highest governmental offices", "and yet in the people itself has escalated a dreadful Jew-hatred, aggravated to the point of being an annihilation will". This development indeed "refutes clearly and cruelly the liberal ideology of progress". Prinz adds further that even "within the Zionist camp", hardly a place that tends to "get its illusions up about the Jewish question", there is no clarity about "the recent development". Actually, it has been "unilaterally reduced to *economic* factors. On the other hand, precisely the specific political connection, i.e., the significance of *democracy* for the configuration of the Jewish question, has been hardly recognized so far" [68] (p. 113).

According to Prinz, the emergence of democracy was bound to bring about some unattainable expectations among Jews. The main causes of the disappointment are not hard to find. For starters, "the antisemitism of the previous ruling class, especially of the court and the nobility, was no doubt appropriate to make us sympathetic to the revolution, all the more so since the working class now pushing for power was partially led by Jews and utterly rejected antisemitism. The memory of the great blessings that we thank the French Revolution for, and the idea that every discrimination of Jews contradicted the idea of democratic equality, did the rest". Finally, "parliamentarianism, 'government by discussion', appeared to be suitable to a high extent to the rationalism and dialectical endowment of Jews" [68] (p. 113).

Now, those who, on this basis, set their hopes so high in democracy "*mistake* for the most part *democracy* for *liberalism*". Indeed, "*Parliamentarianism* itself, which at the time of its introduction among us was already in its way down everywhere, is not essentially a democratic institution, but an institution that belongs in the world of liberalism, relying on the belief in discussion and the public sphere, as Carl Schmitt has made it clear in his outstanding book *The Crisis of Parliamentary Democracy* (2nd ed., 1926)". Thus, "we have now in Germany a difficult, perhaps deathly crisis of parliamentarianism (just like that of liberalism in general), whereas democracy in other, direct forms proves itself to be probably utterly vital" [68] (p. 113).

On account of the highly contingent, almost orthogonal relationship between democracy and liberalism identified by Schmitt, Prinz drives home the point that:

> For us Jews, *the mix-up between the democratic and the liberal idea of equality is of the outmost importance and much more fatal [than for other people].* It is in no way plausible that the "equality of all those who bear a human face" corresponds to the intellectual world of democracy—quite to the contrary! "The equality of all persons as persons", thus explains Carl Schmitt, "is not democracy but a certain kind of liberalism, not a state but an individualistic-humanitarian ethic and worldview". "Every actual democracy rests not only on the principle that only equals are equal, but also on the principle that unequals will not be treated equally". To democratic equality thus belongs a certain *equality of substance*. This can consist of the agreement between religious convictions, like, e.g., the XVII century English sectarians, or in certain physical or moral qualities, like the virtus

of classical democracy. But "since the nineteenth century it has existed above all in membership in a particular nation, in *national homogeneity*". [68] (p. 113)

Finally, Prinz asks himself what results from these considerations vis-à-vis "the knowledge of the German Jewish question". Prinz's answer is that "*democracy does not diminish but strongly increases the political significance of the strangeness and the otherness of single groups of populations*". His conclusion is that "far away from 'solving' the Jewish question in the lands of Galuth [exile], *democracy* on the contrary *exacerbates* it, it raises a series of new and important questions, and partly it creates and partly it exposes a situation that increases the significance of the Zionist way out" [68] (p. 113). In other words, democracy can solve the Jewish question for Jews but only in a country of their own[29].

An anonymous reviewer of the present paper is right on the money in saying that "many Jews rejected Zionism in the 1920s and 1930s". As stated above, whereas German Jews amounted to less than one per cent of the population of Germany, German Zionists, in turn, amounted to no more than four per cent of that one per cent. The vast majority of German Jews still clung to the process of emancipation and assimilation in their country of birth, so much so that they had no doubt whatsoever about their being German citizens of the Jewish faith—and this is why they could not possibly anticipate the ascent of National Socialism. My point, however, is that German political Zionists were as Jews as those who did not share the idea of creating a Jewish state in Israel; as a result, this goes some way towards showing that there is nothing antisemitic (or worse) about the structure and content of the concept of the political.

This same reviewer adds that some German Jews rejected Zionism "as a rhetoric not distinguished enough from National Socialism". Extraordinary as it may sound, there may well be some meeting points between National Socialism and Zionism. For instance, they both agree that the process of emancipation and assimilation of Jews was a failure, but no one would seriously claim this entails that Zionism is National Socialism. In fact, the German Zionists who took a leaf out of Schmitt's concept of the political could not possibly be further removed from National Socialism since—to recall just one of the many differences seen above—according to Schmitt's concept of the political: "*An enemy is not someone who, for some reason or other, must be eliminated and destroyed because he has no value. The enemy is on the same level as am I*".

German Schmittian Zionists also learned firsthand that an open-house, all-inclusive approach to political reasoning would entail welcoming even National Socialism into the house; in other words, they learned that some degree of political exclusion and, therefore, homogeneity is unavoidable. This is the uncomfortable lesson taught by Schmitt's concept of the political that resonated so loudly with some German Zionists.

Naturally, political exclusion is a very dangerous business; we ought to tread very lightly as we make political decisions in the deepest sense of the word, viz., as we engage in exclusionary measures. We must be sure about the legitimacy of exclusion. Otherwise, we run the danger of becoming precisely what we are trying to avoid. However, there seems to be little room for disagreement when it comes to the need to exclude, say, National Socialism from the political game. Exclusion can be said to be quite like emergency brakes in vehicles like trains or subways; although they are meant to help us avoid some grave dangers, they are susceptible to dangerous uses themselves. However, it would be even more dangerous to get rid of them altogether.

To conclude, at a time when the ideas and institutions of liberal democracy are besieged (sometimes literally, as in the quite recent cases of the United States in 2021 and Brazil in 2023) by political movements fundamentally hostile to their assumptions on government and society, for all their praiseworthy inclusionary self-understanding, liberal democracies would do well to take Schmitt's concept of the political—particularly the exclusionary nature of political reasoning—very seriously, and hence they would also do well to mind the fact that they themselves have enemies. In Schmitt's own words, "it would be a deranged calculation to suppose that the enemy could perhaps be touched by the absence of a resistance". Liberal democracies must be aware then of the fact that if they no longer

possess "the energy or the will to maintain themselves in the sphere of the political, the latter will not thereby vanish from the world" [5] (p. 53). Only a weak political discourse and its attendant institutions will disappear.

**Funding:** This research received no external funding.

**Data Availability Statement:** No new data were created, Data sharing is not applicable to this manuscript.

**Conflicts of Interest:** The author declares no conflict of interest.

## Notes

1. I would like to express my gratitude to the four anonymous reviewers of this paper and the Academic Editor of this issue for their very helpful comments and criticisms. They have enabled me to clear up some misunderstandings and to develop the points made in this paper.

2. Having received a letter from Emil Lederer, the editor of the *Archiv für Sozialwissenschaften und Sozialpolitik*, Schmitt considered the possibility of sending what was meant to be a chapter for *Constitutional Theory* as an essay to that journal. This is how the first edition of the essay of 1927 came to light.

3. In the prologue to the 1963 edition of *The Concept of the Political*, Schmitt thanks Julien Freund and George Schwab for having tipped him off about the need to distinguish between different types of enmities. See [6] (pp. 17–18).

4. See [7] (p. 82): "the partisan needs legitimation if he is to be included in the political sphere and not simply sink into the criminal realm".

5. See [6] (p. 10)

6. The irony is that Machiavelli does not employ the term "political" (*politico*) in *The Prince* at all but reserves the term for his republican work. See [11] (p. 810).

7. See [19] (p. 75).

8. In a note, Schmitt added that: "The more profound thinkers of the nineteenth century soon recognized this. In Jacob Burckhardt's *Weltgeschichtliche Betrachtungen* (of the period around 1870) the following sentences are found on 'democracy, i.e., a doctrine nourished by a thousand springs, and varying greatly with the social status of its adherents. Only in one respect was it consistent, namely, in the insatiability of its demand for state control of the individual. Thus *it blurs the boundaries between state and society* and looks to the state for the things that society will most likely refuse to do, while maintaining a permanent condition of argument and change and ultimately vindicating the right to work and subsistence for certain castes'. Burckhardt also correctly noted the inner contradiction of democracy and the liberal constitutional state: 'The state is thus, on the one hand, the realization and expression of the cultural ideas of every party; on the other, merely the visible vestures of civic life and powerful on an *ad hoc* basis only. It should be able to do everything, yet allowed to do nothing. In particular, *it must not defend its existing form in any crisis*—and after all, what men want more than anything is to participate in the exercise of its power. The state's form thus becomes increasingly questionable and its radius of power ever broader'" [5] (pp. 23–24, emphasis added).

9. William Shakespeare, *Coriolanus*, III.1.

10. "Politics within the state, as Carl Schmitt himself pointed out, is political only in a secondary degree, unlike foreign policy, for example. It is public policy [*Polizei*] in the classic sense, care and struggle for good order within and of the community, a politics that does not exceed or explode the pacified framework and its integration within it. Thus, the accomplishment of the state as a political unity is precisely to relativize all the antagonisms, tensions, and conflicts that arise within it, making it possible—within the framework of the state's peaceful order—to debate them, struggle for answers, and eventually arrive at solutions in public discourse and through orderly procedures" [25] (p. 71).

11. The 6 January 2021 attack on the U. S. Capitol by Trump's supporters and the assault on the seats of power in Brasilia in 2023 by Bolsonaro's will surely both ring a bell.

12. Quoted in [30] (216–217). I'm grateful to the anonymous reviewer who kindly reminded me of the fact that, to be sure, it would be simply false to claim that the Weimar Republic was entirely lacking in legal weapons to face up to its enemies. The German Parliament had enacted some statutes in defence of the republic (see [31] (p. 258)). Schmitt himself was obviously aware of the existence of these statutes [see 39] (pp. 26, 113). The problem was that they were not applied against the enemies of the republic due to the predominance of the all-inclusive approach to political and legal reasoning.

13. Quoted in [33] (p. 276).

14. See [5] (p. 36): "Nothing can escape this logical conclusion of the political"; [39] (p. 111) "Politics is unavoidable and ineradicable".

15. If a politics of "us and them" is fascist (see [41]), then any kind of politics is fascist.

16. Most people in liberal democracies prefer to be free and equal within their own nation rather than to be free and equal cosmopolitan citizens because cosmopolitan citizenship would make it hard for them to live and work in their own language and their own liberal culture. The preference for particular nations holds even though it prevents people from having the freedom to

work and vote elsewhere and makes it hard for those who are not citizens of a liberal country to live and work in it. See, e.g., [43] (pp. 93, 95). We tend to forget, e.g., that John Stuart Mill thought that "free institutions are next to impossible in a country made up of different nationalities" (quoted in [44] (p. 98)). Thus, even if liberal democracy were to be the only political show in town or in the world, it would still entail exclusionary consequences.

17    See [6] (p. 102).

18    Although at the turn of 1932, Schmitt still sent his best New Year's wishes to Feuchtwanger, on 12 April 1933, he withdrew *The Concept of the Political* from Duncker & Humblot on the grounds that "Between Arnold Bergsträsser and Gerhard Leibholz [two Jewish authors], it [the *Concept* treatise] appears in a false, distorting light". In November 1933, Feuchtwanger still suggested to Schmitt that "he might act as a point of contact in endeavours to secure a new 'status' for Feuchtwanger's 'comrades in faith and race, these sadly divine unlucky fellows punished by God': 'A Reich commissar working with an absolutely reliable Jewish expert and middle man would be needed for the introduction of such a regulation and as the person responsible for it'" [33] (p. 288). Feuchtwanger appealed to their past friendship, but to no avail. Schmitt never got back to him. Having been imprisoned for six weeks in Dachau as a "protected Jew [*Schutzjude*]" in the context of the November pogrom of 1938, Feuchtwanger finally decided to leave Germany for England and was able to do so before the outbreak of the war. After the war, he returned to Germany as a translator for the American army. He also taught re-education courses to German prisoners of war in England, where he died on 17 July 1947 (see [33] (pp. 288–289)).

19    In her quite recent book, *The Emotional Life of Populism*, Ella Illouz claims that "Israel displays what Carl Schmitt defined as the essence of 'the political'—that is, the distinction between friend and enemy", without meaning this to be a compliment. In the same book, she refers further to Schmitt as "the Nazi legal scholar who joined the party in 1933" without any further qualification, as though Schmitt's character or decisions would simply overlap with his entire work. See [53] (pp. 24, 158).

20    Schmitt is usually referred to as the "crown jurist". Although the term was originally employed to describe his role as advisor to the last government of the Weimar Republic, it became an epithet meant to portray his endorsement of National Socialism, at least during the first years of the new regime—in spite of the fact that the real "crown jurist" of National Socialism was Hans Frank.

21    Günter Maschke explains that "Hobbes did not possess a good name in the Reich of brown Jacobins, for Hobbes is above all a statist" ([18] (p. 195)). In the introduction to his monograph on Hobbes of 1938, Schmitt says to be quite "aware of the danger implicit in the subject. *Stat nominis umbra*. The name leviathan throws a long shadow; it has fallen on the work of Thomas Hobbes and will in all likelihood also fall on this little book". Actually, Schmitt draws the book to a close by identifying himself with Thomas Hobbes: "Today we grasp the undiminished force of his polemics, understand the intrinsic honesty of his thinking, and admire the imperturbable spirit who fearlessly thought through man's existential anguish, and, as a true πρόμαχος [champion], destroyed the murky distinctions of indirect powers. To us he is thus the true teacher of a great political existence; lonely as every pioneer; misunderstood as is everyone whose political thought does not gain acceptance among his own people; unrewarded, as one who opened a gate through which others marched on; and yet in the immortal community of the great scholars of the ages, 'a sole retriever of an ancient prudence'. Across the centuries we reach out to him: *Non jam frustra doces, Thomas Hobbes!* [Thomas Hobbes, now you do not teach in vain!]" [17] (pp. 2, 86). In his review of the book, the Nazi jurist Otto Koellreutter complained about the fact that "a 'Hobbes renaissance' does not belong in the political thought of our age", because of Hobbes's main claims: "*Auctoritas, non veritas facit legem*. The power of government is the only lawgiver, law is only the positive law laid down by the state" (quoted in [18] (p. 196)).

22    Raymond Aron also reminds us that the Promethean ambition "is one of the intellectual origins of totalitarianism" [58] (p. 199).

23    See [33] (p. 480).

24    It seems then that Pinhas Rosen (formerly Fritz Rosenblutt), Israel's Minister of Justice in 1948, was hardly the first Zionist to be interested in the work of Carl Schmitt as he consulted Schmitt's *Constitutional Theory* at the time of the preparation of the draft of the Israeli Constitution of 1948 (see [62] (p. 99)). According to Christian Linder, in Schmitt's later age, his "masochistic hate for the Jews had finally oozed out of him, save for the occasional outbreak well into the 1970s". The Jewish question came up again one evening around 1970 in Pasel, a village on the outskirts of Plettenberg, where Schmitt's family had been relocated. Rüdiger Altmann, a student from Marburg who went on to be a well-known publisher, exclaimed: "Herr Professor, I will not take part in this discussion. You dedicated your 'constitutional theory' to your Jewish friend, Fritz Eisler, who fell in World War I for Germany. This ought to prohibit you from speaking in such a way here". Schmitt fell silent. Ernst Hüsmert agrees that by that time, "Schmitt had finally made his peace with the Jews because ever since they 'got a piece of land under their feet and possessed their own state, they started behaving like every other nation'" [54] (p. 165).

25    In fact, Bernstein's book was written in 1923 and published in 1926, before the first edition of *The Concept of the Political* (1927), so Schmitt himself could also be said to have answered vividly to Bernstein's book, and not just the other way around. In a letter, Bernstein tried to persuade Schmitt to write a review of his book but to no avail.

26    See Bernard M. S. van Praag, "Introduction" in [66] (p. x).

27    Naturally, this passage brings to mind the last paragraph of section 3 of *The Concept of the Political*, the first edition of which was written in 1927: "Nothing can escape this logical conclusion of the political. If pacifist hostility toward war were so strong as to drive pacifists into a war with nonpacifists, in a war against war, that would prove that pacifism truly possesses political energy because it is sufficiently strong to group men according to friend and enemy. If, in fact, the will to abolish war is so strong that it

no longer shuns war, then it has become a political motive, i.e., it affirms, even if only as an extreme possibility, war and even the reason for war. Presently this appears to be a peculiar way of justifying wars. The war is then considered to constitute the absolute last war of humanity. Such a war is necessarily unusually intense and inhuman because, by transcending the limits of the political framework, it simultaneously degrades the enemy into moral and other categories and is forced to make of him a monster that must not only be defeated but also utterly destroyed. In other words, he is an enemy who no longer must be compelled to retreat into his orders only" [5] (p. 36).

28  In the prologue to the English version, written after the creation of the state of Israel, Bernstein revealingly holds that "the Jews *were* everywhere weak and defenseless minorities and the most tempting object for an outburst of passionate hate which prefers, understandably enough, discharge without risk of retaliation or defense" [66] (p. 20, emphasis added).

29  This was exactly the point made by Theodor Herzl in *The Jewish State*: "We are one people—our enemies make us one without our will, as it has always been in history. (...). We are one people, one people. We have honestly endeavored everywhere to dissolve ourselves in the surrounding community of peoples and only to preserve the faith of our fathers. We are not allowed to do so. In vain are we loyal patriots, and in some places even exuberant patriots; in vain do we deliver the same sacrifices of life and property as our fellow-citizens; in vain do we exert ourselves to increase the glory of our countries in the arts and sciences, and the wealth of our countries by trade and commerce. In our countries, where we have already lived for centuries, we are still cried down as strangers, often by those whose ancestors were not yet in the land where our fathers had already been sighing. Who the stranger is, that can be decided by the majority; it is a question of power, as everything in the relations between nations" [69] (pp. 38, 27). This goes some way towards explaining the apparently awkward remark made by Herzl in his diary: "The anti-Semites are right. If we grant them that, then we too will be happy" (Theodor Herzl's Diaries, Book I, 17 June 1895, quoted in [70] (p. vi). Precisely, "If we compare the portrait of the Jew drawn by anti-Semitism in literature with the iconography of the diaspora Jew cultivated by Zionism, the difference is rather slight" [71] (p. 108). So, Herzl's claim seems to refer to the fact that Jews themselves were mainly to blame for being considered eternal foreigners and nomads for their tardiness in reentering the political sphere: it was about time for them to get back in the saddle.

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
