# Peer review of "It Had to Be You: Carl Schmitt on Exclusion and Political Reasoning"

_philosophies, doi:10.3390/philosophies9020048_

Round 1
Reviewer 1 Report
Comments and Suggestions for Authors
I read through the manuscript with some care, but I’m not sure how I should qualify it – whether it is a scientific paper or an erudite illustration of a few Schmittian concepts.
The first half looks to me as a theoretical discussion – not so much a criticism, let alone a justification – of Schmitt’s concept of the political. Here the author introduces the principle of “formative exclusion”, as s/he names it, to reject the widespread view that Schmitt’s political theory pivots on enmity. I’m very sympathetic to the idea that the political in Schmitt is not a polemogenic principle, as it is first and foremost meant to create amity between friends. However, I don’t think this case can be made only based on the principle of formative exclusion. It implies a sort of self-justifying omnis determinatio est negatio which calls for theoretical vindication. Yet, the fact that something is practically difficult does not imply that this very thing is theoretically impossible.
The main problem of the manuscript, though, has to do with the second half, which makes the article more of a collection of considerations and quotations from Schmitt’s various texts (never contextualized) as well as from other authors. The exposition certainly doesn’t shape up as a structured argument raising specific claims. It covers various topics more or less connected to Schmitt’s concept of the political that are exposed and briefly discussed, most often by relying on Schmitt’s own words. As I pointed out, Schmitt’s texts are never contextualized as they should, and are rather used as evidence for what the author says – although one always exposes oneself to the allegation of cherry-picking when one adopts such an argumentative strategy.
An illustrative example is the section on Schmitt’s antisemitism. The author first reports a statement by Edgar Feuchtwanger, son of Ludwig Feuchtwanger, a Jewish friend of Schmitt’s. Then a long quote ensues. Then it is stated that Schmitt could not be antisemite before 1933 because “Ludwig Feuchtwanger would not have embraced them [Schmitt’s pre-1933 texts] with open arms, Schmitt would had been hardly a favourite author of his” – and in all evidence this is hardly a robust case. Then the author admits that “there is no beating about the bush with the fact that for all his rejection of revolution, Schmitt ended up collaborating with the Nazi regime”. But shortly thereafter, the author quotes political scientist Waldemar Gurian to remind the reader that Schmitt “had Jewish friends and admired Jewish scholars”. Then again, a further reference to another renowned Schmitt scholar, George Schwab, who is presented as “a Jewish Holocaust survivor” to say, I assume, that his pro-Schmitt testimony cannot be discounted as sneakily antisemite. Then the author quotes Schmitt saying that Zionists understood his concept of the political much better than others. Then an example “the way in which some German Zionists were taken by 706 Schmitt’s work”. And the other anecdotes, stories, quotations, and so on.
In sum, one never comes across a vindication of the principle of “formative exclusion” introduced on p. 2 of the manuscript – which would have made the manuscript a scientific paper. But the second half, too, is unconvincing. It is not a reasoned biographical sketch, nor is it a methodologically-informed historical study. There’s certainly a good wealth of material that is nicely marshalled for the paper to be an interesting read, but as one goes through it, one gets carried here and there with no visible blueprint.
The author could restructure and refocus it by developing the first half and by providing a theoretically sound illustration and, most importantly, justification of the principle of formative exclusion. However, this would make it a totally different article.
Based on this, my advice is that the article be rejected.
A final suggestion, should the editors decide to ask for revisions, is for the author to remove the many colloquialisms that abound in the manuscript.
Reviewer 2 Report
Comments and Suggestions for Authors
The author of this paper (A) makes a huge effort to sanitise Schmitt. That Zionists may admire Schmitt is far from showing that his concept of the political and of the friend/enemy distinction is not anti-liberal in an extreme way, as there have always been Zionists, and this is even more true today, who embrace the correct view of Schmitt, which A ignores, that the enemy is someone who is existentially 'other' and who must be destroyed in order to achieve the substantive homogeneity of the Volk. Such Zionists simply replace the Jew with the Arab. A also ignores the fact that Schmitt was a virulent antisemite, as shown in his diaries both pre and post war.
A's choice of literature is either wilfully blind or worse. For example, A cites Ernst Forsthoff, a work of 1931, but neglects to mention that Forsthoff was a Nazi by conviction as well as a disciple of Schmitt and his essay on formalism in public law illustrates the antisemitism latent in Schmitt's legal theory, as does Schmitt's essay, also not cited by A, on the Jewish influence in German law.
A discusses Schmitt on the role of a judge but does not mention Kelsen's response which showed that Schmitt was engaging in a political act of charlatanism in his claims about who is the guardian of the constitution.
I could go on. But while A is clearly smart and knows a lot about Schmitt, that this is the case just lowers my estimation of this essay as it increases the sense that it is an exercise in sanitation.
Reviewer 3 Report
Comments and Suggestions for Authors
This submission seeks to make the strong argument that Carl Schmitt's 1926 argument about the concept of the political (really about the essence of politics) in fact cannot be reconciled with his full-throated endorsement of Nazism between 1933-36. This is an interesting argument, and the author moves toward making it in this essay. In its present form, though, I do not consider it ready for publication. First, there are some more superficial reasons not to accept the manuscript now. Second, there are several more substantive reasons to question whether it's ready for the public, not least the question of whether the author adequately and systematically investigates the core question raised. Responding to both, I believe, would result in a much shorter submission--which is good, since the current submission is in fact quite long for a journal article.
First, the formal issues. The author has a lot to say--which is what tends to happen when Schmitt is discovered, one issue leads to another! But overall, the text seems to meander from one topic to another rather than to develop tight points in support of the main argument. This tendency to slide from one topic into another is accompanied by the tendency to bring in long quotations from Schmitt and from others--which are often left under analyzed. These quotations are often not clearly indicated by quotation marks, as well, a tendency that grows stronger in the second half of the paper: if I cannot see clearly what is the author's language, what is being quoted, or even clearly who is being quoted, my job as reader is made much more difficult. The final section on Zionism doesn't feel complete, indeed seems to just break off. In short, this was not ready for submission. Last not least, the citation system neither conforms to humanities nor author-date format and is not easy to use.
Second, and more important, the substantive issues. I think that the author gets a LOT right here. Especially the first third or so of the essay is really important, where the author is reconstructing Schmitt's defense of the friend-enemy distinction as essential to understanding a polity. That friend-enemy distinction operates first and foremost vis-a-vis the external world: a state exists as an armed collective, the "enemy" is the external and the not-self. That's what sovereignty is about. This is an argument from the "essence" of politics--which lies in the international realm--that fits within a "realist" conception of international relations and ultimately as well in the (politically disastrous) trajectory of German international law doctrine, which essentially denied international law from the point of view of state sovereignty. Schmitt's strong argument about how the enemy is the opponent, not the evil to be annihilated, is also relevant although not necessarily that well connected with the realities of warfare. At any rate, it can certainly be read as a defense of an IR realism that seeks to avoid demonizing the enemy in foreign affairs--as a "humanizing" rhetoric even if it rejects "humanitarian" notions of international law that (allegedly) lead to dehumanizing practices.
The concomitant of this notion of sovereignty and distinction is a "neutralization" of the state internally, which grants those within the state rights as participants in the state, whether as civil actors or as political actors. Herein lies Hobbes.
But then there is a peculiar shift that the author enacts, replicating Schmitt, but without noting very well: from neutralization to intensification. The Spinozian/Hobbesian position is, as Schmitt wrote in the late 1930s (after his allegedly momentary involvement with Nazism, and not really articulated by the author), essentially liberal, protecting a core freedom of conscience; Schmitt's position seems different, asserting a substantive unity of the constitution. Undermining or attacking that substantive unity--perhaps, Schmitt suggests in a quote used by the author, by affirming godlessness--makes the internal citizen into an enemy of the state, which ceases to be neutral. This suggests not the neutralization of the state but its intensification. I don't see a careful discussion of this apparent paradox here: in fact Schmitt is affirmed as a kind of Hobbesian liberal while the author's summary seems to lead in a different direction. Is militant democracy really the same concept of the political as the one pointing toward the other outside of the borders of the sovereign state?
Which takes us to a second, related point. Schmitt is often taken as the author of the theory of militant democracy, according to which democracies have to act against internal enemies but have historically failed to do so. There is a great amount of literature on this topic, but I'm not seeing it in the current contribution. First, is it really true that Weimar failed to defend itself? I thought that myth had been destroyed. Many different measures were taken to protect the republic (broadly grouped under Republikschutz). Some of these probably worked and saved the republic for a time. Others were used by Nazis and Communists to defame the republic and to argue that it was hypocritical. BUT THE MEASURES TO PROTECT THE STATE AGAINST INTERNAL ENEMIES EXISTED. It was concrete actors on the right who undermined the measures taken against the Storm Troopers in 1932, for example. Second, is it really true that militant democracy honestly defends democracy? Loewenstein's ideas served also to justify the radical suppression of internal critics during the Cold War. Is that what the author wants to defend? The discussion of militant democracy by people like Müller addresses this issue. Third, this suppression of the internal enemy according to Schmitt involves defense of a substantive, "core" constitution prior to the written constitution (the mere Rechtsstaat), as the author suggests. So who decides what these basic values are? Schmitt comes to focus on property and family. Why? Both are limited in basic ways in the constitution itself; is Schmitt cherry-picking by declaring some parts of the constitution essential and others inessential? I am not telling the author what to argue here, but I am just missing his digging deeper into the paradoxes of the problems that his own strong argument about Schmitt's notion of politics.
Which leads to another issue. At one point in the paper the author repeats Schmitt's sarcastic rejection of the argument that a qualified majority can revise the constitution (Art 76 of the constitution). But let's assume that all constitutions need to leave a way open to revision. If the decision by a qualified majority (which is, yes, an act of power) is not legitimate to enact a revision, what is? Not, according to the author and Schmitt, a decision by a judge. So by whom, then? By a president, who might well be elected by less than a majority of the voters according to Weimar law? Democracy is actually about power, as the founders of Weimar knew. They did not disagree with Schmitt about the necessity of power. But they did not accept that there was some entity above the people (God, Leader, Army, etc.) who should have the power. That point is implied but not fully brought forward by the author.
Related, Schmitt himself changes his positions over time, and this change over time seems lost in the current article. In 1912, Schmitt insisted that judges could not be restrained by statute. In 1928 and 1932, he insisted that they had to be restrained to the letter of the law, i.e. by statute. In 1934-36, he insisted that judges could operate in the interest of a higher law above the statute, but restrained by the sense of the Volk. That's just one example.
I am not arguing that the author is wrong, I am rather arguing that the author needs to be a lot more focused in this essay. The author has a point that needs to be made very concrete. The concept of the political formulated in 1926-1928 is very specific and specifies the inevitability of a friend/enemy distinction grounding the republic, which also implies an internal set of values that have to be affirmed--assumed to be prior to the constitution. Develop this with reference to the strongest arguments in agreement with the thesis (Christian Meier, Ernst-Wolfgang Böckenförde, neither cited); note that the "concept of the political" changes in other contexts (here Meier is key in showing the 1933-34 transition; the work on partisans after 1960 meanwhile comes from a different context, which distanced Schmitt for a second time from a friend like Forsthoff). The author need not defend all of Schmitt in order to make a focused argument. The more focused the better.
I am not convinced that the section on Zionism works. It forgets just how many Jews rejected Zionism in the 1920s and 1930s as a rhetoric not distinguished enough from National Socialism (see Klemperer's comments in his diaries from the time!). It also sounds more than a little apologetic, especially given that the author doesn't quote Schmitt's diary entries, his entries in the Glossarium, or his texts from the mid-1930s on Jews. The author is making a clear and useful argument; this muddies the water and makes the useful argument carry the aftertaste of an apologetic exercise, which will undermine the reception of the big argument!
So my suggestion is NOT that I disagree, but that to have an impact this essay needs to be revised pretty dramatically. It seems to me it should limit its breadth to its big argument--and then carry out that big argument with more care (for example actually taking into consideration Schmitt's arguments during the Nazi phase of his career, 1933-38).
Comments on the Quality of English Language
overall good but in need of editing.
Reviewer 4 Report
Comments and Suggestions for Authors
The essay is original, well-written and thought-provoking. It is a very good contribution to Schmitt-scholarship as well as to the topic at hand, the notion of exclusion in politics. It is also very rich in content, abounds with historical details and engages very well with an extensive secondary literature.
The examination of Schmitt's position about exclusion, especially in the famous essay The Concept of the Political, is well-balanced and correct, as are the conclusions. The part on the Jewish intellectuals' reaction and, generally speaking, Schmitt's relationship with Judaism is fair.
I have only one concern/suggestion: the very richness of the essay somewhat blurs the line of argument. The author should add a conclusion, perhaps less brilliant than the current one and more didactic, in which he/she clearly states Schmitt's main argument about exclusion, its consistency with a liberal order, and its obvious relevance for today's imperilled liberal democracies.
Round 2
Reviewer 1 Report
Comments and Suggestions for Authors
I appreciate that the author is trying to address the major flaws I've identified, but the changes s/he has made certainly don't make the article publishable.
As an example, I'd like to comment on the last part added to this new version. The author writes that our times are dangerous and that liberals would do well to take Schmitt's politics seriously because they have enemies of their own. But I wonder, is this the crux of Schmitt's critique: "Wake up, the enemy is at the gates"? I don't think so. Rather, he was making the point that liberals play the political while denying that the political is what it is. So, liberalism does not qualify as a political theory because it tries to "sterilise" the political and yet cannot avoid its exclusionary nature. It excludes while denying that it does so. Then the author writes: " Liberal democracies must be aware then of the fact that if they no longer possess ‘the energy or the will to maintain themselves in the sphere of the political'". Is this a sociological or a political statement? What is the evidence for it?
All in all, I'm afraid the author has missed the point of the review. In her/his reply, s/he explains why s/he could not do what I would have liked to see in the paper. But this is no justification for the shortcomings I have highlighted. The author adds that “the reason why I did not discuss the general context of Schmitt’s work was, again, that in order to economize space, I preferred to focus on the structure of Schmitt’s political thought, his descriptive and normative claims”. Yet the article can hardly be regarded as a detailed analysis of Schmitt's descriptive and normative claims. It is mainly a restatement, with many references to anecdotes as well as opinions about Schmitt.
Basically, everything I wrote in my review still applies to the current version.
My suggestion on colloquialism was ignored. But English is fine.
Reviewer 4 Report
Comments and Suggestions for Authors
No special comment. I believe the original submission was good, this revised version is clearer in the line of argument and has an appropriate conclusion.
Round 3
Reviewer 1 Report
Comments and Suggestions for Authors
The revised manuscript can be published as is.